# Synergistic olfactory processing for social plasticity in desert locusts

Inga Petelski[1,2,3,5], Yannick Günzel [1,2,3,4,5] ✉, Sercan Sayin[2,4], Susanne Kraus[2] & Einat Couzin-Fuchs [2,3,4] ✉

Desert locust plagues threaten the food security of millions. Central to their formation is crowding-induced plasticity, with social phenotypes changing from cryptic (solitarious) to swarming (gregarious). Here, we elucidate the implications of this transition on foraging decisions and corresponding neural circuits. We use behavioral experiments and Bayesian modeling to decompose the multi-modal facets of foraging, revealing olfactory social cues as critical. To this end, we investigate how corresponding odors are encoded in the locust olfactory system using in-vivo calcium imaging. We discover crowding-dependent synergistic interactions between food-related and social odors distributed across stable combinatorial response maps. The observed synergy was specific to the gregarious phase and manifested in distinct odor response motifs. Our results suggest a crowding-induced modulation of the locust olfactory system that enhances food detection in swarms. Overall, we demonstrate how linking sensory adaptations to behaviorally relevant tasks can improve our understanding of social modulation in non-model organisms.

The ability to adapt to a changing social environment is a fundamental aspect of life. Locusts exhibit a distinct ability to switch between a solitary and gregarious lifestyle rapidly, rendering them an ideal model for studying social plasticity and the emergence of collective behavior. Furthermore, these transitions are central to the formation of large-scale destructive locust outbreaks, which are precipitated by an autocatalytic process of positive feedback between population density and aggregation behavior[1–4]. When population density spontaneously increases, formerly solitarious locusts are forced to conglomerate on limited resources, leading to further increases in local density and subsequent directed aggregation of animals that have become gregarious. These dynamics also trigger changes in other density-dependent traits, including metabolism, developmental, and reproductive physiology, which together foster population growth and swarm formation (cumulatively termed 'phase change',[3]). As crowding further increases, the demand for nutrients rises, compelling large locust aggregations to leave areas with low food availability and forage despite associated risks of predation and cannibalism[5–9]. These rapid density-dependent adaptations pose questions about the neural processing in changing social environments that allow animals to adjust their decision-making appropriately in a context-dependent manner.

The transition to group living also alters the sensory information available to locusts when searching and selecting feeding sites. Gregarious desert locusts (*Schistocerca gregaria*) effectively utilize both asocial (e.g., sight and/or smell of food) and social (e.g., presence or action of conspecifics) cues for their foraging decisions[10]. Specifically, the decision to join a food patch is positively influenced by the number of conspecifics on the patch, as well as by its quality and the locust's prior experience with it. However, our understanding of the sensory processes that mediate these density-dependent decisions remains limited.

Like many animals, locusts use olfactory information to locate food, find mates, and avoid predators or toxins. Invariably, olfactory systems face the challenge of encoding and decoding a vast array of

[1]International Max Planck Research School for Quantitative Behavior, Ecology and Evolution from lab to field, 78464 Konstanz, Germany. [2]Department of Biology, University of Konstanz, 78464 Konstanz, Germany. [3]Department of Collective Behavior, Max Planck Institute of Animal Behavior, 78464 Konstanz, Germany. [4]Centre for the Advanced Study of Collective Behaviour, University of Konstanz, 78464 Konstanz, Germany. [5]These authors contributed equally: Inga Petelski, Yannick Günzel. ✉e-mail: yannick.guenzel@uni-konstanz.de; einat.couzin@uni-konstanz.de

chemical structures across varying odor concentrations and mixtures in turbulent environments. Knowledge gained about olfactory systems across the animal kingdom has demonstrated a largely typical organization of a single neuropile—the antennal lobe (AL) in insects and the olfactory bulb in mammals—located only one synapse away from the peripheral sensory neurons[11]. To meet the challenge of complexity in chemical space, the antennal lobe (like the olfactory bulb) acts as an early processing center in which numerous incoming olfactory receptor neurons (ORNs) converge on fewer output projection neurons (PNs). Non-linear summation in such synapses and gain control mediated by local neurons enables the coding of odors over dynamic ranges[12].

Invertebrate and vertebrate model systems in olfactory research, including rodents[13], zebrafish[14], and fruit flies[15], have a typical early olfactory coding architecture. Convergent input and output neurons form a computational unit called the glomerulus. Each glomerulus receives input from all ORNs expressing the same olfactory receptor type, and most output neurons are exclusive to a particular glomerulus[11]. Such a glomerular structure effectively organizes olfactory coding in linear and parallel channels. In locusts, however, the AL structure and connectivity deviate from those of many insects[16]. In contrast to, for example, 50 individually identified glomeruli in the fruit fly[15], it houses an unusually large number of more than a thousand so-called microglomeruli and more strikingly, all output PNs innervate multiple microglomeruli (no uniglomerular PNs have been described to date), and each ORN projects to multiple glomeruli[16–20]. Consequently, it is largely unknown how this structural difference impacts locust olfaction and if it is related to the notable phenotypic plasticity many locust species exhibit.

While the wiring of the locust antennal lobe remains unresolved, research on odor processing in locusts has played a pioneering role in advancing our understanding of combinatorial coding in olfactory systems[21–23]. Using intracellular and extracellular recording techniques and computational models, locust odor representation has been postulated as a dynamic combinatorial code[19,20], suitable for rapid discrimination of odor identity and intensity[24,25], as well as for contrast enhancement when stimuli overlap or follow distractor odors[26–28]. Despite the importance of combinatorial coding and ensemble dynamics, most data sets captured only a fraction of antennal lobe output at a given recording or lacked the cell's identity, which prevented assigning structure to function. Both problems can be addressed with functional imaging techniques that allow monitoring of network-spanning dynamics, which is particularly important for the highly distributed innervation of the locust AL.

To this end, we have established a functional imaging protocol to investigate the spatial organization of odor-evoked activity in a large subset of PNs of the locust AL. We map the interaction between food and social odors in gregarious and solitarious locusts to examine how odor representation is impacted by crowding. We link our findings to the animals' preferences in a behavioral assay by estimating the contribution of different sensory cues to foraging decisions. Our results reveal a crowding-induced olfactory modulation and advance the accessibility of locusts as a model system for studying collective behavior and the neuronal mechanisms underlying social plasticity.

## Results

### Social plasticity and the sensory basis of foraging decisions

We start our investigation of how population density is linked with sensory processing and decision-making by analyzing the contribution of different sensory modalities to foraging decisions, focusing on social and food olfactory cues. For this, we employed simple patch selection assays that presented locusts with four options: food (blackberry leaves, denoted as *Lvs*), a social setting (eight gregarious locusts, *Lct*), a combination of leaves and locusts (*LvsLct*), and an empty control patch (*Ctr*). We controlled the availability of olfactory

and visual cues by using different stimulus containers that either provided both visual and olfactory cues or limited one of them (see Fig. 1a and Methods for details). We observed that gregarious desert locusts, *Schistocerca gregaria*, exhibited a strong attraction to the patch containing both leaves and locusts, as shown in Fig. 1b (*Vis*(+) *Olf*(+)). This attraction diminished in the absence of either visual (*Vis*(−)*Olf*(+)) or olfactory cues (*Vis*(+)*Olf*(−), see also Supplementary Fig. 1 for animal trajectories). In contrast, solitarious locusts typically disregarded social stimuli and predominantly approached patches where food was presented alone (Fig. 1d). Notably, just as with gregarious animals, the choices of solitarious locusts were most pronounced when both visual and olfactory cues were available.

We dissected the contributions of different sensory cues to the decision process via a Bayesian decision-making model[10,29], extended to encompass the four types of cues available in the current assay: vision, olfaction, social, and food. Specifically, we estimated the reliability of each information class—socio-visual, food-visual, socio-olfactory, and food-olfactory—to elucidate how these factors influence locusts' choices (Fig. 1c and e, light gray bars; reliability parameter $C_i$ in Eqn. (1)). The close alignment between our model's predictions and the observed choices (light and dark gray bars respectively in Fig. 1b and d) suggests that the integration of available sensory cues can effectively describe locust decisions. This model enabled us to dissect further and quantify the parameters that contributed to these decisions. Notably, we found that gregarious animals were highly responsive to social cues, with the highest information reliability scores associated with the presence of conspecific odors at a patch (Fig. 1c). In contrast, solitarious-reared animals appeared deterred by the sight of conspecifics, as indicated by their negative scores in this context (Fig. 1e).

### Functional imaging revealed differences in olfactory processing between gregarious and solitarious locusts

Building on the importance of olfaction in patch discrimination, we mapped how respective odor cues are represented in the locust antennal lobe (AL). An anatomical reconstruction of the AL, using backfill labeling of olfactory receptor neurons (ORNs) and projection neurons (PNs), shows the arrangement of PN cell bodies on the anterior surface of the AL, with their dendrites situated beneath, encircling a central fiber core. Here, they form functional units, microglomeruli, with ORN axon terminals (Fig. 2a–c). To monitor local calcium signals as a proxy for neuronal activity, we established a protocol for functional imaging via mass retrograde labeling of PNs with the fluorescent calcium indicator Cal-520[30]. This technique was optimized to label the vast majority of PNs, facilitating subsequent optical recordings (see Methods for further details).

Furthermore, we developed a data-driven analysis pipeline (Fig. 2d) that segments the highly arborized structure of the locust AL into distinct activity granules and overcomes the limitations of biased manual selection of specific regions of interest. Calcium signals were captured at focal depths of $35 \pm 25\ \mu m$ and $140 \pm 25\ \mu m$, chosen to maximize the number of detectable cell bodies and glomeruli, respectively (see Fig. 2c for anatomical reference with examples of functional maps below).

Interested in the AL representation of the cues presented in our behavioral experiment, we conducted a chemical analysis to identify dominant volatiles in the headspace of the blackberry leaves. Gas chromatography-mass spectroscopic analysis featured two prominent appetitive plant volatiles[19]: leaf alcohol acetate (cis-3-Hexenyl-acetate, denoted as *Laa*) and the leaf alcohol 3-Hexel-1-ol (denoted as *Z3hl*), Fig. 2e. We then used the identified volatiles and an extraction of the locust odor (*Lct*) as stimuli to measure PN responses in gregarious and solitarious animals.

Each stimulus elicited a response apparent as calcium signals in a large set of PNs distributed across the AL (functional maps at the focal depth of the cell bodies show odor-induced responses in regions

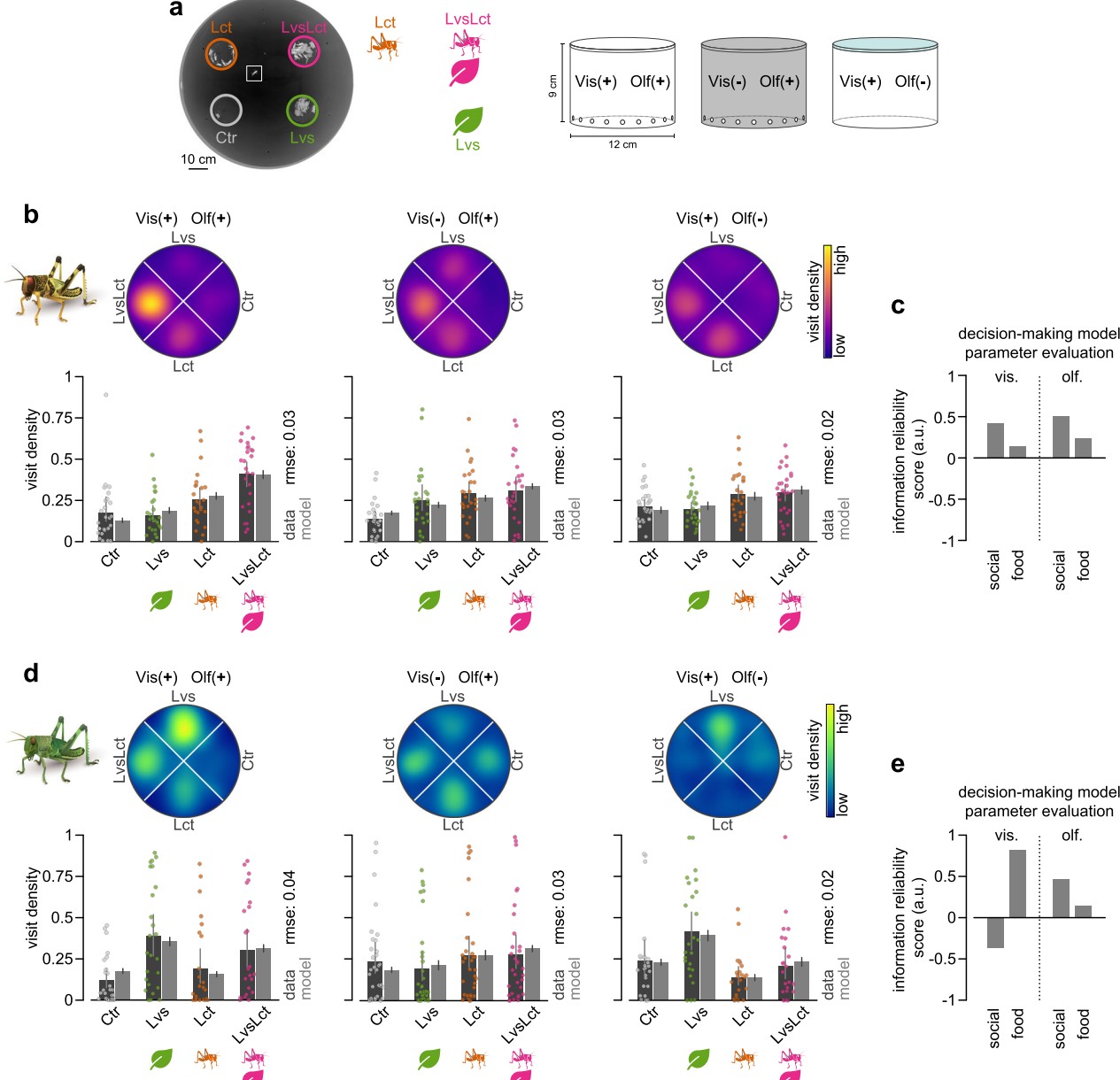

**Fig. 1 | Interactions between social and food cues in patch selection. a** Animals were tested in a circular arena with four patch options (blackberry leaves *Lvs*, locusts *Lct*, leaves and locusts *LvsLct*, and control *Ctr*) under one out of three sensory conditions (visual and olfactory cues available: *Vis*(+)*Olf*(+); olfactory but no visual cues: *Vis*(-)*Olf*(+); visual but no olfactory cues: *Vis*(+)*Olf*(-)) to isolate the effects of visual and olfactory cues during patch selection. On the left, we show a color-inverted, example frame from a gregarious animal trial. Here, the super-imposed circles indicate the four patches, and the square shows the location of the locust. We used different stimulus containers for the different sensory conditions (schematics on the right) that were either transparent and odor-permeable with surrounding holes to provide both visual and olfactory cues (*Vis*(+)*Olf*(+)), opaque with holes to restrict visual cues (*Vis*(-)*Olf*(+)), or transparent but sealed with Par-afilm (blue-shaded top) to restrict olfactory cues (*Vis*(+)*Olf*(-)). **b** We tested gre-garious locusts under the three sensory conditions. Heatmaps depict average

locust density in each stimulus quarter (blackberry leaves *Lvs*, locusts *Lct*, leaves and locusts *LvsLct*, and control *Ctr*) for different sensory conditions (visual and olfactory cues available: *Vis*(+)*Olf*(+), left (*n* = 26); olfactory but no visual cues: *Vis*(-) *Olf*(+), center (*n* = 26); visual but no olfactory cues: *Vis*(+)*Olf*(-), right (*n* = 28)). Swarm plots represent individual locusts' mean density in each stimulus quarter, dark gray bar plots with error bars indicate corresponding grand means with 95% confidence intervals, and light gray bar plots with error bars indicate means and 95% credible intervals from a decision-making model (5000 simulations; see Methods). **c** Information reliability scores for each information class (social, food) and sensory modality (visual, olfactory), were obtained by fitting a decision-making model to the patch selection data in (**b**). **d, e** Similar experiments and analysis as in (**b, c**) but for solitarious animals (*Vis*(+)*Olf*(+): *n* = 25; *Vis*(-)*Olf*(+): *n* = 32; *Vis*(+)*Olf*(-): *n* = 25). Source data are provided as a Source Data file.

matching in size, shape, and number to individual cell bodies; exam-ples in Fig. 2f and h). In both gregarious and solitarious locusts, response magnitudes to the leaf odor (the prevalent leaf component *Laa*) were higher than those to the locust odor *Lct* (green and orange curves in Fig. 2f and h show population response averages for *Laa* and *Lct*, with respective responses to the solvent as control in gray).

Notably, however, was the respective change in response magnitude when both odors were simultaneously delivered as a mixture (*LaaLct*; containing half the amount of each single component; see Methods for mixture preparation). PN responses of gregarious animals significantly increased when adding the social component to the food-related one (*p* < 0.0001; paired two-tailed bootstrap randomization test), while in

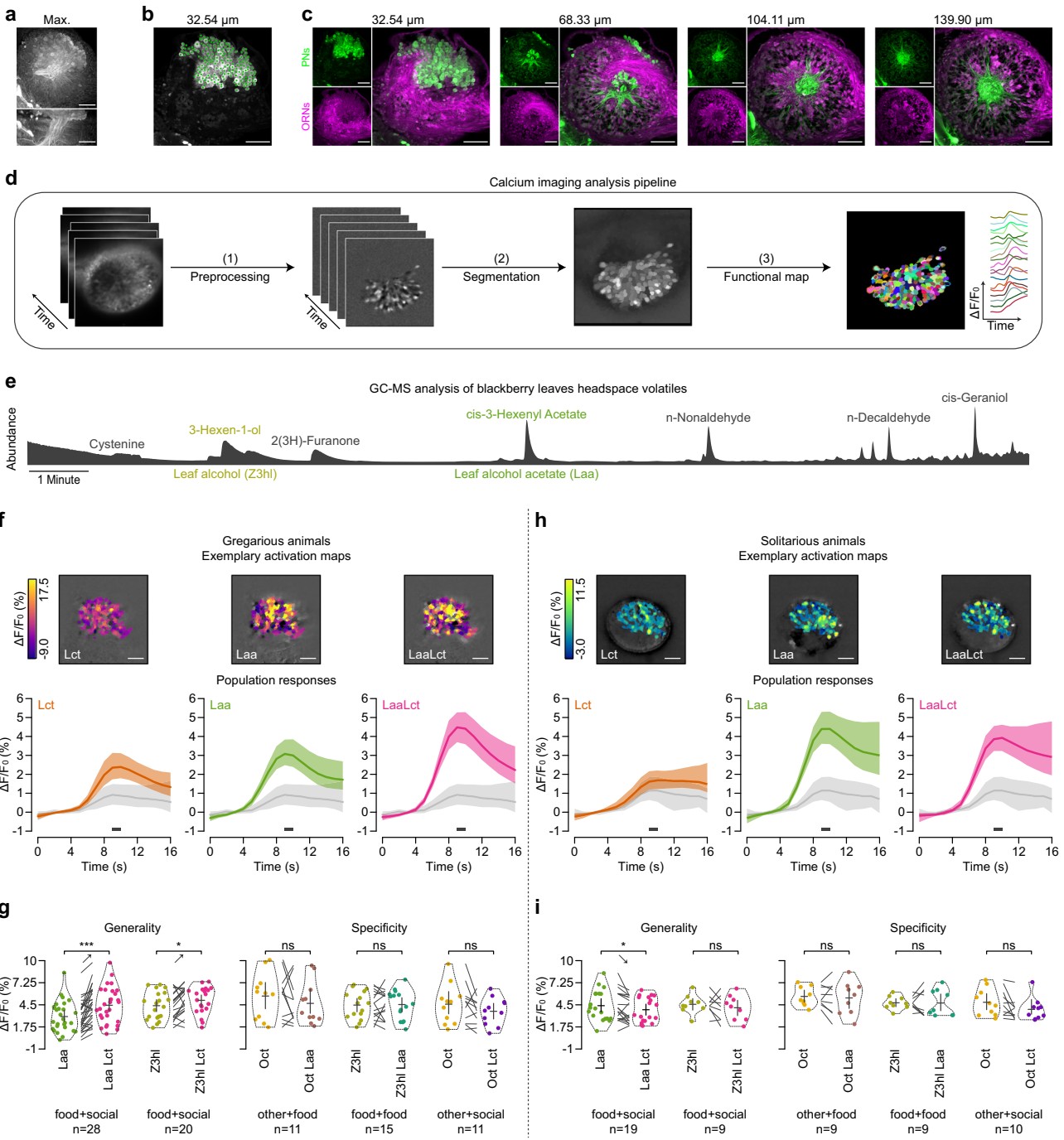

solitarious animals, adding the social component resulted even in a significant decrease in response magnitude ($p = 0.0316$; see average time courses in Fig. 2f–h with the respective comparisons for individual data points in the left panels of Fig. 2g–i for gregarious and solitarious animals, respectively). This difference is also reflected by the population response trajectories (Supplementary Fig. 4), illustrating distinct response patterns for the different stimuli that are generally consistent across the two phenotypes, albeit with noticeable differences (especially in the first principal component).

We observed similar findings when testing PN responses to crushed blackberry leaves (*Lvs*) in combination with the locust odor (Supplementary Methods and Supplementary Fig. 2a–e). Here, the response magnitude to *LvsLct* mixture was significantly higher compared to the individual components *Lvs* and *Lct* in gregarious ($p = 0.02$; paired two-tailed bootstrap randomization test) but not in solitarious

($p > 0.99$) locusts. Since this experiment was conducted with a wide-field microscope, which—despite being faster—lacks the capacity for thin optical slices, we replicated the experiments with leaf alcohol acetate *Laa* for a direct comparison (Supplementary Fig. 2g–j). With similar response magnitudes and comparable differences between the phenotypes, these results align with the confocal data. Additional tests of antennal responsiveness to the odors via electroantennograms (EAG), reflecting summed receptor potentials along the antenna[31], show almost identical response patterns in gregarious and solitarious antennae (Supplementary Fig. 3c, d). This observation suggests the differences between the social phenotypes are not present presynaptically to the AL. Altogether, we propose that neural processing of odor mixtures in locusts may be subjected to a density-dependent modulation in the AL, highlighting a potential neural mechanism underlying differences in social behavior.

**Fig. 2 | Functional imaging of the antennal lobe representation of the odor cues used in patch choice experiments. a** Maximum projection of a confocal stack of a projection neuron (PN) backfill staining along the z-axis (top) or the y-axis (bottom). **b** Identification of 181 individually stained PN somata (superimposed green dots) in one section of the scanned locust antennal lobe. **c** Confocal scans with a double backfill staining of PNs (green) and olfactory receptor neurons (ORNs, magenta) at various imaging depths (indicated above; $Z = 0$ corresponds to the anterior surface of the AL). **d** Schematic diagram detailing the staining protocol for PNs in locust antennal lobes for in-vivo functional calcium imaging and the subsequent analysis pipeline (see Methods for specifics). **e** Gas chromatography analysis identifies leaf alcohol acetate (*Laa*, cis-3-Hexenyl Acetate) and leaf alcohol (*Z3hl*, 3-Hexen-1-ol) as prominent food-related volatiles in blackberry leaves. **f** Confocal microscopy data of odor-induced responses to the locust odor *Lct* (left), leaf alcohol acetate *Laa* (center), and their combination *LaaLct* (right). Average activity time courses (grand means with 95% confidence intervals as shaded areas) show the responses of all gregarious animals ($n = 28$), while the false color-coded images show the mean intensity projections of an exemplary animal during the window for analysis (dark gray bars below the time courses; stimulus onset is indicated by vertical dotted lines). Responses to the solvent are shown as light gray time courses. **g** Evaluation of gregarious animals' mixture responses to test for generality (increase in response magnitude for other food-related stimuli upon addition of the social component) and specificity (the increase in mixture response occurs only for the combination of a food-related with a social stimulus). Swarm plots show individual animal means during the window for analysis, violin plots indicate probability density estimates, while crosses depict grand means and 95% confidence intervals for all stimuli (*Laa*, *Z3hl*, and 1-Octanol *Oct*) and respective combinations (generality: $n_{(food1+social)} = 28$, $n_{(food2+social)} = 20$; specificity: $n_{(other+food)} = 11$, $n_{(food+food)} = 25$, $n_{(other+social)} = 11$). **h, i** Correspond to (**f, g**), but for solitarious animals (generality: $n_{(food1+social)} = 19$, $n_{(food2+social)} = 9$; specificity: $n_{(other+food)} = 9$, $n_{(food+food)} = 9$, $n_{(other+social)} = 10$). Statistical inferences in (**g**) and (**i**) were based on paired two-tailed bootstrap randomization tests (ns: not significant; *$p < 0.05$; **$p < 0.01$; ***$p < 0.001$ with gregarious $p_{(food1+social)} = 3.58*10^{-5}$, $p_{(food2+social)} = 0.0237$, $p_{(other+food)} = 0.2781$, $p_{(food+food)} = 0.8534$, $p_{(other+social)} = 0.1411$; solitarious $p_{(food1+social)} = 0.0316$, $p_{(food2+social)} = 0.3539$, $p_{(other+food)} = 0.6247$, $p_{(food+food)} = 0.9531$, $p_{(other+social)} = 0.0601$). Scale bars: 100 μm. Source data are provided as a Source Data file.

## Generality and specificity of synergistic mixture response

Next, we tested whether the increased response to the mixture could be generalized to other food-related components. For that, a subset of animals was tested with additional odor combinations, beginning with the second appetitive plant volatile in the blackberry leaf headspace, *Z3hl*[32]. As for *Laa* and crushed leaves, PN responses to *Z3hl* in gregarious locusts significantly increased when the social component was added ($p = 0.0237$ for *Z3hl* vs *Z3hlLct*; paired two-tailed bootstrap randomization test). At the same time, in solitarious locusts, the response to the same mixture was not higher than to the individual components ($p = 0.3539$).

Considering the consistent phenotypic differences across all mixtures tested so far, we explored the possibility that mixture processing is broadly different in the two social phenotypes. We addressed this by evaluating whether the observed increase in gregarious response magnitude is specific to a mixture of a social with a food-related stimulus or whether it can also be observed for other mixtures. For this, we tested additional odor combinations containing a non-social and/or non-food component, combining equal amounts of a neutral (non-food and non-social) component 1-Octanol with *Laa* (*OctLaa*, other+food), *Laa* with the second leaf component *Z3hl* (*LaaZ3hl*, food+food) and the mixture of 1-Octanol with *Lct* (*OctLct*, other+social). For gregarious locusts (Fig. 2g), neither of the combinations—other+food, food+food, other+social—induced a change in PN responses magnitude ($p = 0.2781$, $p = 0.8534$, and $p = 0.1411$, respectively; paired two-tailed bootstrap randomization tests). Similarly, but as expected, no increase was observed for solitarious animals either ($p = 0.6247$, $p = 0.9531$, and $p = 0.0601$). Taken together, these findings suggest that the consistently increased response observed in gregarious locusts when combining a social with a food-related odor— *LaaLct*, *Z3hlLct*, and *LvsLct* (Supplementary Fig. 2 for the later)— represents a phenotype-specific difference in which the salience of food odors is heightened when accompanied by the presence of conspecifics.

## Combinatorial odor processing with mixture-specific PNs in the locust antennal lobe

We further analyzed the odor-induced response maps to understand better how the mixtures are processed and what may mediate the increase in gregarious animals' response magnitude. At first, we noted that responses to the different stimuli were highly overlapping. Most PNs responded to multiple tested odors, with only a few responding to a narrower subset (Fig. 3a). Interestingly, out of these narrower ones, we could detect mixture-specific PNs which were activated by a mixture but not by its components, showing a clear phenotypic difference in the prevalence of *LaaLct*-specific cells

(red and blue bars in Fig. 3b show the proportion of mixture-specific PNs in gregarious and solitarious locusts; $p_{(LaaLctonly)} = 0.0220$, $p_{(OctLaaonly)} = 0.9284$, $p_{(LaaZ3hlonly)} = 0.1161$, $p_{(OctLctonly)} = 0.5295$; two-sample two-tailed bootstrap randomization tests).

To investigate these PN response patterns, we employed a unified analysis approach. For this, we projected the response maps from all PNs and all tested odors across all animals into a unified odor response space, as illustrated in Fig. 3c. Despite the considerable overlap described above, we discerned distinct odor response vector classes. These classes aided in delineating PN response types, which we used to explore potential differences between the social phenotypes (Fig. 3d, and Fig. 3c for the linear discriminant analysis 2-D projection). Upon evaluating these response vectors, we found that the population response to a specific odor primarily drives certain vectors. For instance, vector 1 is predominantly influenced by *Lct*, and similarly, vector 2 by *Z3hl*, vector 4 by *Laa*, and vector 5 by *Oct*. Conversely, the other vectors encapsulate a more combinatorial response profile. For example, vector 3 demonstrates a combinatorial pattern with responses to a broad set of stimuli. The distribution of PN types according to the odor space clustering was largely similar in both phenotypes, with a difference in prevalence mainly for *Lct*-positive and *Lct*-negative types (pie charts in Fig. 3d show the proportion of each PN type in red and blue for gregarious and solitarious locusts, respectively).

## Combinatorial odor maps stay consistent across multiple repetitions

Previous studies on locust odor mapping have shown that, although the population-level responses reliably determine odor identity, there can be substantial variability at the level of individual PNs[25]. Thus, we continued our investigation by examining how consistent odor responses are across repetitions. Response consistency was determined for three general odors in the PN cell bodies (similar optical slice as before) and their dendritic terminals (concentrically arranged glomeruli captured at the focal depth that maximizes the number of visible glomeruli, 140 μm). As expected from a combinatorial coding system, odor-induced responses had considerable overlap (displayed as a color-coded overlay in Fig. 4a). However, these responses were distinct across different odors and remained consistent for the same odor over multiple trials (see Fig. 4b and Supplementary Fig. 5 for individual trials in the example animal, and Fig. 4c for the population response).

## Phase-dependent differences are consistent across cell compartments

By recording responses to social and food-related odors in the PN dendrites of gregarious (Fig. 4d–f) and solitarious (Fig. 4g–i)

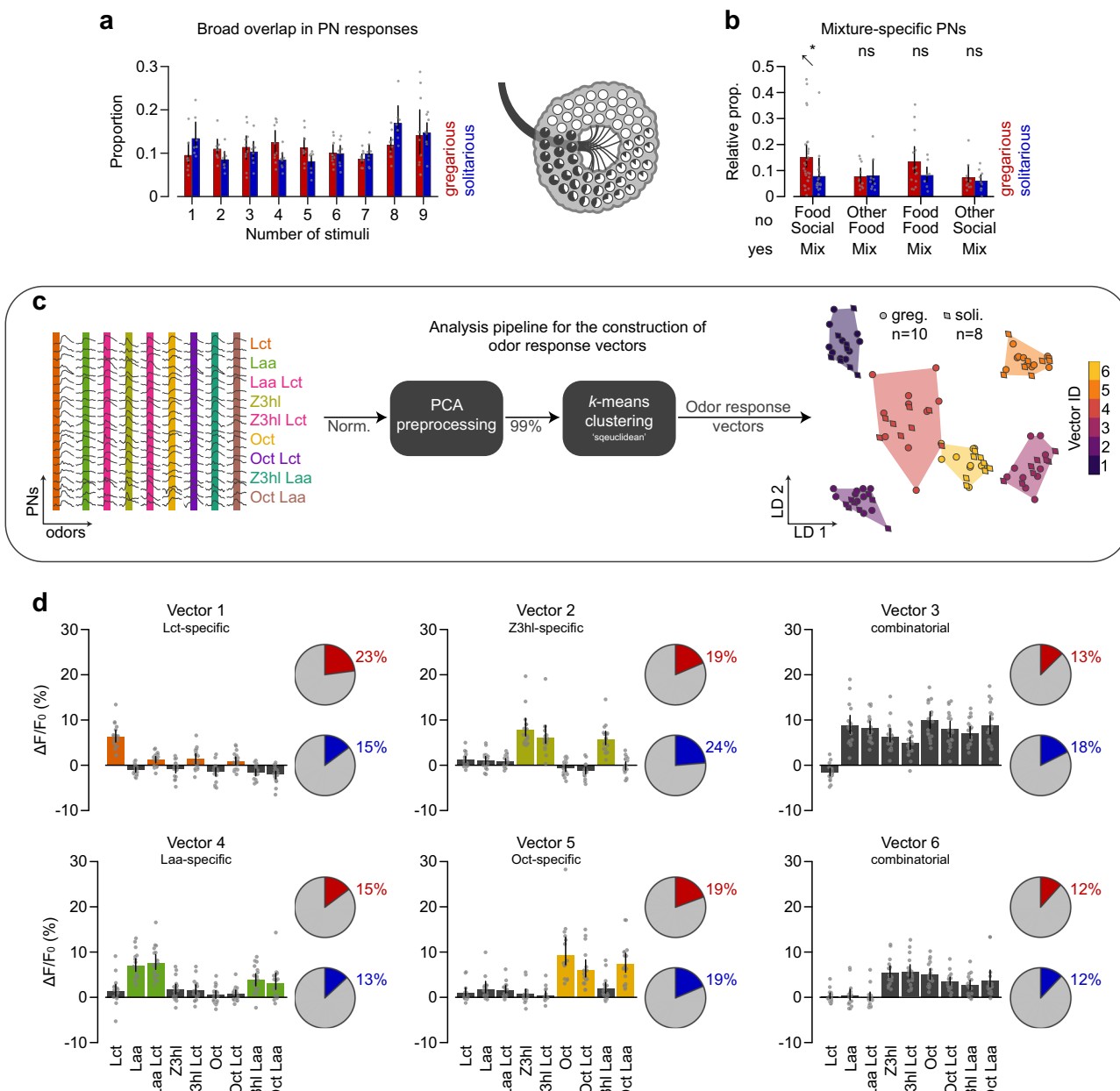

**Fig. 3 | Combinatorial coding of odor responses in antennal lobe projection neurons of gregarious and solitarious animals. a** The proportion of gregarious (red) and solitarious (blue) PNs that respond to a given number of stimuli, as symbolized on the right by the pie charts with the proportion of responses for each PN, overlaid onto the antennal lobe illustration. All animals ($n_{(greg.)}$ = 10; $n_{(soli.)}$ = 8) were tested with nine stimuli (locust odor *Lct*, leaf alcohol acetate *Laa*, leaf alcohol *Z3hl*, 1-Octanol *Oct*, as well as their mixtures *LaaLct*, *Z3hlLct*, *OctLaa*, *Z3hlLaa*, and *OctLct*). **b** The relative proportion of mixture-specific PNs − regions that are responsive to a two-component mixture ('yes') but not to its two components ('no'). The proportion of mixture-specific PNs was calculated out of all PNs responding to that mixture (specific+non-specific) for gregarious (red; $n_{(LaaLctonly)}$ = 28, $n_{(OctLaaonly)}$ = 11, $n_{(Z3hlLctonly)}$ = 15, $n_{(OctLctonly)}$ = 11) and solitarious (blue; $n_{(LaaLctonly)}$ = 19, $n_{(OctLaaonly)}$ = 9, $n_{(Z3hlLaaonly)}$ = 9, $n_{(OctLctonly)}$ = 10) animals. **c** Diagram

explaining the clustering of individual PNs' response magnitude sets into odor response vectors. The scatter plot displays the linear discriminant analysis 2-D projection after clustering to visualize the classification into activity vectors (circles: gregarious ($n$ = 10); diamonds: soliatrious ($n$ = 8) animals). **d** Representation of each odor response vector along with pie charts displaying the relative allocation of gregarious (red) and solitarious (blue) PNs to the respective vector. Bars, color-coded to signify response vector type (color: odor-specific; gray: combinatorial), indicate grand means with 95% confidence interval error bars ($n_{(greg.)}$ = 10; $n_{(soli.)}$ = 8). Statistical inference in (**b**) was based on two-sided two-tailed bootstrap randomization tests (ns: not significant; \**p* < 0.05: \*\**p* < 0.01: \*\*\**p* < 0.001: with $p_{(LaaLctonly)}$ = 0.022, $p_{(OctLaaonly)}$ = 0.9284, $n_{(Z3hlLaaonly)}$ = 0.1161, $n_{(OctLctonly)}$ = 0.5295). Source data are provided as a Source Data file.

animals, we noted a trend further corroborating what we observed in the cell bodies and widefield imaging above. In gregarious animals, the response magnitude to the mixture (*LaaLct*) was again significantly higher than to *Laa* alone ($p$ = 0.0118; two-tailed bootstrap randomization test). In contrast, we observed a strong trend for a decreased mixture response in solitarious

animals ($p$ = 0.0761). This is in line with a larger proportion of *LaaLct*-specific glomeruli in gregarious (0.12 [0.08, 0.18], mean [95% confidence interval]; Supplementary Table 1) than in solitarious animals (0.05 [0.02, 0.08]), and more *Laa*-specific glomeruli in solitarious (0.23 [0.11, 0.48]) than in gregarious (0.06 [0.04, 0.11]) animals.

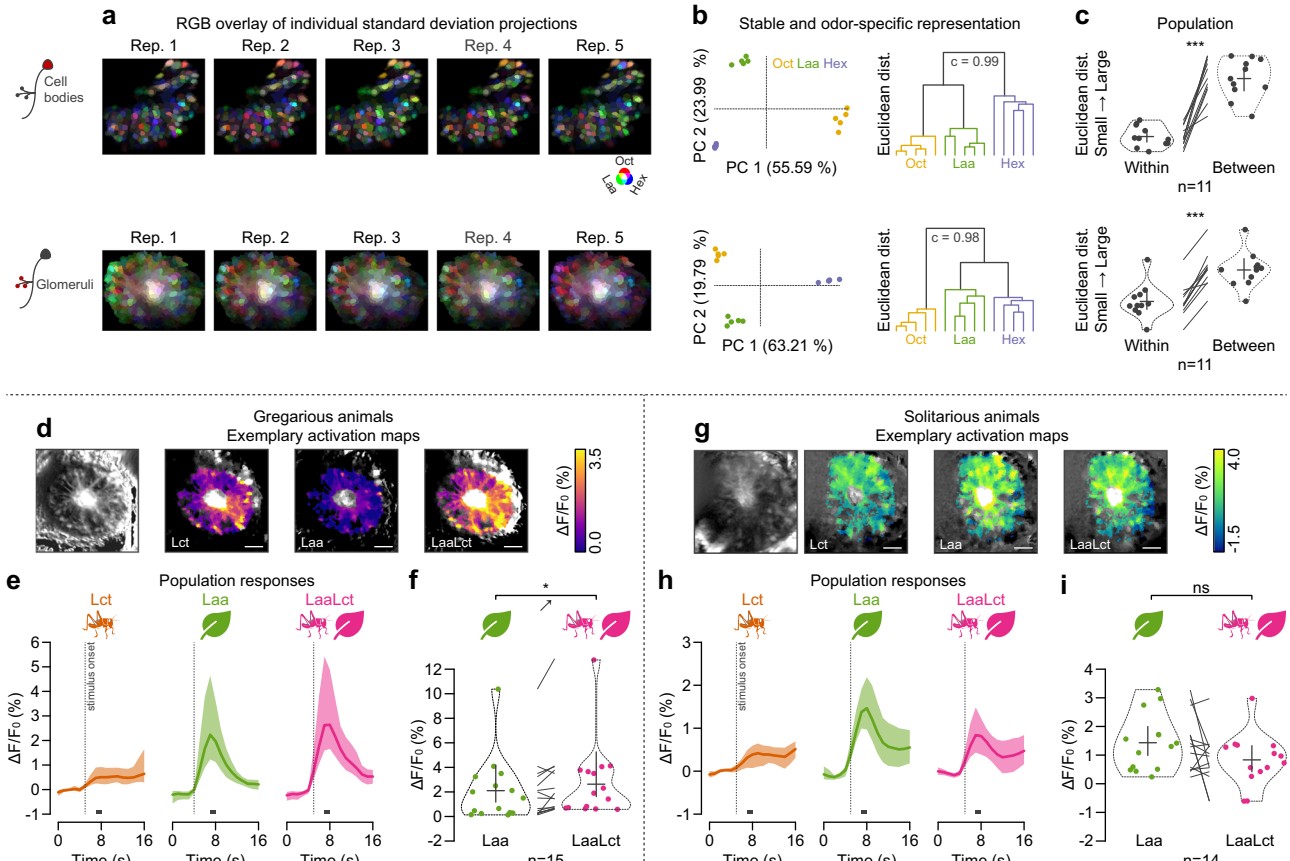

**Fig. 4 | Consistency of odor response patterns in projection neuron cell bodies and glomeruli. a** Examination of projection neuron (PN) response maps in cell bodies (top row) and glomeruli (bottom row) following repeated presentations of three single-molecule odorants (1-Octanol *Oct*, leaf alcohol acetate *Laa*, 1-Hexanol *Hex*) in an exemplary animal. Individual standard deviation projections of the repeated presentation of three stimuli were overlayed in RGB color space (red: *Oct*, green: *Laa*, blue: *Hex*) to visualize regions responding to one stimulus alone or to several stimuli (see Supplementary Fig. 5 for all standard deviation projections). **b** Principal component analysis (PCA) of PN responses to five trials of each odor with an accompanying hierarchical binary cluster tree dendrogram (**c**, cophenetic correlation coefficient). Data correspond to (**a**). **c** Euclidean distance in PC space between responses of trials with the same odor (within) and different odors (between) in $n = 11$ gregarious animals. Swarm plots show individual animal means, violin plots for probability density estimates, and crosses for grand means and 95% confidence intervals (with $p = 4.46 * 10^{-6}$ for cell bodies and $p = 6 * 10^{-8}$ for

glomeruli). **d** Functional confocal microscopy data of odor-induced responses to the locust odor *Lct*, leaf alcohol acetate *Laa*, and their combination *LaaLct* in the glomerulus layer of an exemplary gregarious animal. The heatmaps show the standard deviation projections across all stimuli (grayscale), alongside the mean intensity projections (color) during the window of analysis (see gray bars in (**e**)). **e** Average activity time courses of all glomerular regions (medial tract excluded; stimulus onset is indicated by vertical dotted lines) across all $n = 15$ gregarious animals (grand means with shaded 95% confidence intervals). **f** Swarm plots of individual animal means during the window of analysis. Violin plots indicate probability density estimates, and crosses show grand means and 95% confidence intervals ($p = 0.0177$, same animals as in (**e**)). **g–i** Correspond to (**d–f**), but for $n = 14$ solitarious animals with $p = 0.1141$. Statistical inference in (**c**), (**f**), and (**i**) was based on paired two-tailed bootstrap randomization tests (ns: not significant; *$p < 0.05$: **$p < 0.01$; ***$p < 0.001$). Scale bars: 100 µm. Source data are provided as a Source Data file.

## Response motif dynamics in the locust antennal lobe reliably predict social phenotypes

Our results repeatedly suggest that social cues synergize with appetitive olfactory signals in antennal lobe projection neurons of gregarious locusts. Moreover, we observed sustained somatic activity patterns (Fig. 2f and h), which hint at complex response profiles following a network-spanning integration. Unlike transient glomerular responses (Fig. 4e and h), which could be more critical for stimulus detection, the sustained somatic patterns may provide a continuous assessment of the olfactory landscape for adaptive behavioral output. In view of that, we continued to characterize individual PN response profiles on the level of the cell bodies to uncover what distinguishes gregarious and solitarious phenotypes.

We applied an unsupervised analysis to all PN cell body response profiles resulting from the three stimuli (*Lct*, *Laa*, *LaaLct*), identifying six distinct odor-induced response motifs (Fig. 5a, b). Ordered by peak magnitude and timing, these motifs categorize PN responses as inhibitory (below baseline motifs 1 and 2), delayed response (motif 3),

excitatory phasic (motifs 4), strong excitatory phasic (motifs 5), and excitatory sustained (motif 6). Examining the proportion of PNs in each response category, which were averaged across animals for each stimulus (Fig. 5c), revealed key differences between gregarious and solitarious locusts (Fig. 5d). Notably, solitarious animals exhibited a higher prevalence of the inhibitory (motif 2) and delayed (motif 3) responses for the locust colony odor, and a higher prevalence of the strong phasic (motif 5) and sustained (motif 6) response for the food-related stimulus. In contrast, gregarious animals exhibited a higher prevalence of the excitatory phasic response (motif 4) for the mixture.

Particularly interested in what underlies the difference in mixture processing between gregarious and solitarious animals, we conducted a deeper analysis of the synergistic responses to the mixture. We tracked how PN response motif assignments change with odors and hence assigned each PN a response motif triplet. For instance, [2|4|5] (Fig. 5e, left) corresponds to the neuron's response to the three presented stimuli (*Lct*, *Laa*, and *LaaLct*). Based on this information, we calculated an individual mixture-interaction score (Bliss score, Fig. 5e,

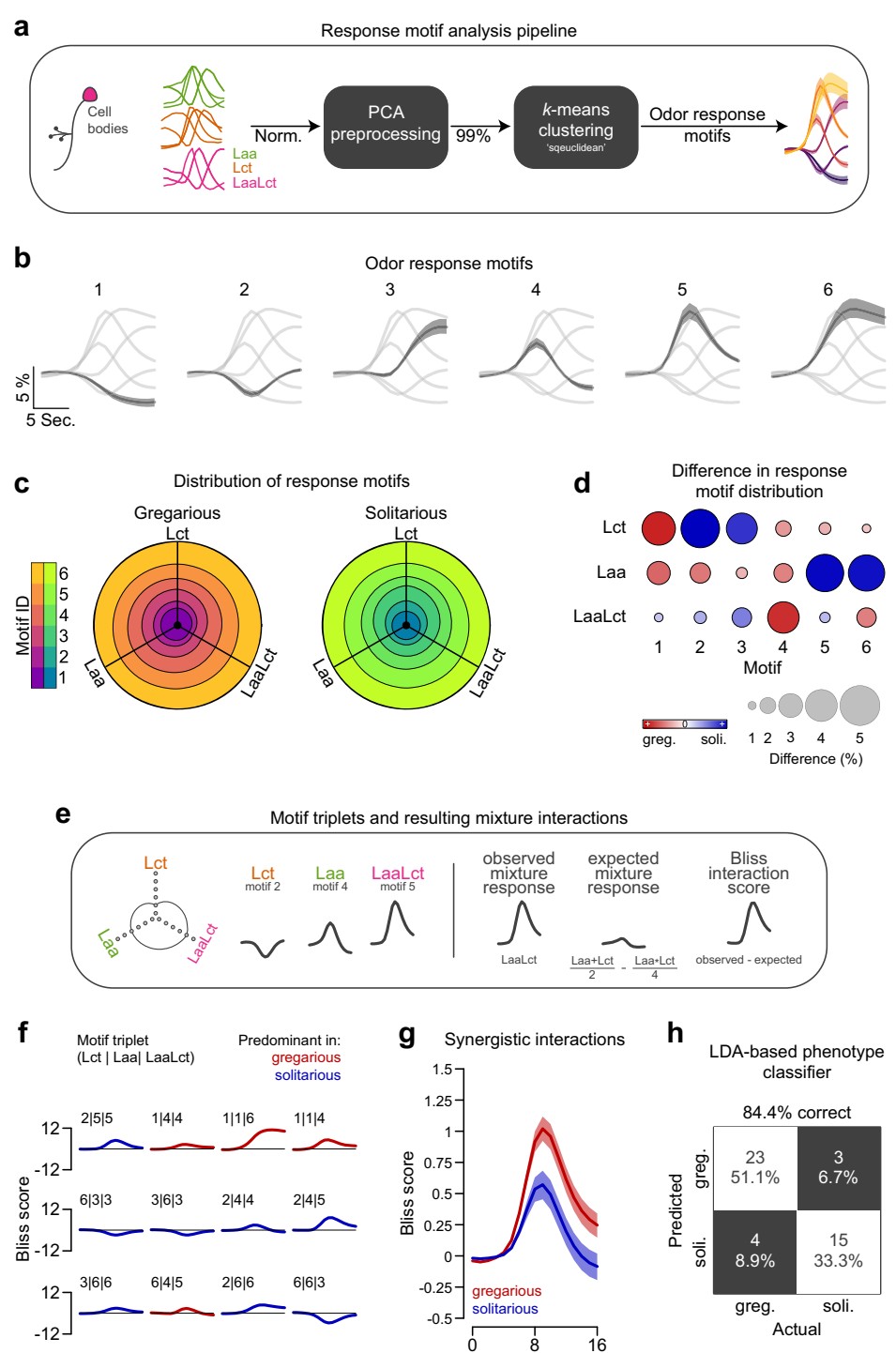

**Fig. a** Response motif analysis pipeline

**b** Odor response motifs

**c** Distribution of response motifs

**d** Difference in response motif distribution

**e** Motif triplets and resulting mixture interactions

**f** Motif triplet (Lct | Laa| LaaLct)   Predominant in: gregarious / solitarious

**g** Synergistic interactions

**h** LDA-based phenotype classifier

right) using Eqn. (3)[33]. Specifically, we estimated the mixture response as expected if there was no interaction between the individual components (*Laa* and *Lct*; Eqn. (3), $f_{expected}$) and compared it for each point in time with the observed response motif to the mixture (*LaaLct*). Consequently, the resulting score distinguishes synergistic interactions (score > zero) from antagonistic ones (score < zero) between *Laa* and *Lct*. We observed that certain motif triplets, and thus certain Bliss interaction scores, were more prevalent in either gregarious or solitarious animals. We identified the 12 triplets that most substantially differentiated the two phenotypes (Fig. 5f; color-coded according to the predominant phase) to further understand the crowding-induced synergy in processing food and social mixtures. Seven of these

combinations involved PNs whose mixture responses were determined by either the stronger (e.g., 2|5|5) or the weaker component (e.g., 3|6|3). The remaining triplets display substantial mutual synergistic (e.g., 1|1|4 and 1|1|6) and antagonistic (e.g., 6|6|3) interactions, which are predominantly found in gregarious (red bliss score profiles) and solitarious (blue) locusts (see also Supplementary Table 2 for triplet prevalence). These types of gregarious synergistic triplets could account for the mixture-specific PNs labeled in Fig. 3b. Further, the proportion of strongly synergistic PNs (Bliss score larger than one standard deviation of the whole distribution) was significantly larger in gregarious (0.25 [0.21, 0.30], grand mean [95% confidence interval]) than in solitarious (0.13 [0.10, 0.20]) animals ($p = 0.0005$; two-sample

**Fig. 5 | Analysis of dynamical odor response motifs. a** Schematic illustration of the clustering process used to categorize the activity profiles of individual cell bodies into six distinct odor-induced response motifs. **b** Temporal profiles of the response motifs, arranged according to peak magnitude and timing. Shown are grand means for each motif across both gregarious and solitarious animals with shaded 95% confidence intervals ($n_{(greg.)} = 28$; $n_{(soli.)} = 19$). **c** Circular stacked bar plots represent the proportion of each response motif for each of the three stimuli (the locust odor *Lct*, leaf alcohol acetate *Laa*, and their combination *LaaLct*). Gregarious animals are shown on the left ($n = 28$), and solitarious animals on the right ($n = 19$). **d** Distinct differences in response motif allocation between gregarious and solitarious animals. Variations in circle color and size denote the degree of disparity (absolute difference in proportion for each motif and odor). **e** Diagram demonstrating the dynamics of motif transitions with odors, which lead to a unique motif triplet for every cell body. We used this to calculate a mixture interaction (Bliss) score for each cell, by comparing the observed *LaaLct* response motif to the ones

expected from the *Lct* and *Laa* components if there was no interaction between them. **f** Analysis of motif interactions using Bliss interaction score time courses to distinguish between synergistic (scores > zero) and antagonistic (scores < zero) interactions between *Laa* and *Lct*. Shown are Bliss profiles of the 12 triplet combinations that exhibited the largest differences in prevalence between gregarious and solitarious locusts (cf. Supplementary Table 1). **g** The average (mean with shaded 95% confidence intervals) Bliss interaction score time courses based on all (but those with identical response motifs across all three stimuli) gregarious (red; $n = 28$) and solitarious (blue; $n = 19$) motif triplets. **h** A confusion matrix showing correct (white) and incorrect (dark gray) phenotype classifications using a linear discriminant analysis model. The model was trained on the dynamics of motif transitions (the proportion of PNs that switch motifs between for a given change between stimuli; see Supplementary Fig. 6) for predicting each animal's phenotype (see Methods for details). Source data are provided as a Source Data file.

two-tailed bootstrap randomization test). Pooling all PN Bliss interaction scores, except those with identical response motifs for all three stimuli, highlights the differences in mixture interactions between the two phenotypes in the context of their response motifs (Fig. 5g).

The increased level of synergy in gregarious processing, together with phenotype-specific motif triplets, suggests that the dynamics of motif transitions between odors carry information about odor processing that may classify the phenotypes. Our comprehensive dataset allowed us to estimate the probabilities of motifs changing, given a change in stimulus (see Supplementary Fig. 6 for heatmaps summarizing the occurrence of all motif transitions). Based on this, we examined the predictive strength of a linear discriminant analysis (LDA) classification model trained on gregarious and solitarious motif transition maps (see Methods for details). This analysis revealed that antennal lobe response dynamics can reliably distinguish gregarious and solitarious phenotypes with 84.4% correct predictions (Fig. 5h). Taken together, our analysis suggests that network interaction variations are manifested in shifts of single PN response profiles, which conjointly give rise to a crowding-induced synergy in the olfactory processing of food and social odors.

## Discussion

We investigated the neural processing of olfactory information in changing social environments, allowing locusts to make decisions appropriately in a context-dependent manner. We established an *in-vivo* functional calcium imaging protocol to reliably monitor the activity of a large subset of antennal lobe (AL) projection neurons (PN). Our results indicate a crowding-induced synergy in the processing of food and social odors that originates from phenotype-specific transition dynamics between distinct response motifs. As a direct consequence, this allowed us to use the AL network dynamics to reliably predict whether a locust was gregarious (group-living) or solitarious. Moreover, our study draws connections between the physiological observations and behaviorally tested patch preferences, providing further insight into the locust decision-making system guiding foraging.

During foraging, the presence of others can indicate both resource quality and location, offering graded and discrete information, respectively[34]. In gregarious locusts, this social information serves as an attractive cue for individuals to join a food patch[10,35], potentially even more than the sight/smell of the food itself (Fig. 1b). We used a Bayesian decision-making rule[10,29], a powerful approach for disentangling multiple contributing factors in behavioral data[36–38], to estimate the relative contribution of the different cue types. Our analysis indicates that both visual and olfactory cues influenced patch selection, and it revealed phenotype-specific differences: while in solitarious locusts visual cues qualitatively predict patch choices on their own (similar preference in *Vis*(+)*Olf*(-) and *Vis*(+)*Olf*(+), Fig. 1d), it is only when both modalities are simultaneously present that a clear

preference is demonstrated in gregarious locusts. Taken together, this revealed the nuanced roles different sensory modalities play in affecting locust foraging decisions, allowing for an investigation of the underlying neural mechanisms.

Given the importance of olfaction for patch selection, we monitored odor-induced neuronal activity across the primary olfactory center. The results showed that PN responses to appetitive odors are modulated by phenotype (i.e., with phase change) when mixed with the social odor. Previous work on gregarious and solitarious locusts described that the two phenotypes have a generally similar AL anatomy (except for a higher number of sensilla in solitarious individuals, potentially relating to their longer developmental period[39]), similar PN spiking properties[20], and response probabilities to single tested compounds[40]. The ability to map odor responses across a large set of PNs in a single animal verified that the phenotypic differences are indeed subtle (i.e., same response motifs in Fig. 5b; largely overlapping response vectors in Fig. 3d; similar response trajectories in an unifying principal component space shown in Supplementary Fig. 4), but clearly distinguish the evaluation of the food+social mixture. Given additionally that electroantennography recordings do not reflect the same phenotypic difference observed in the AL (see Supplementary Fig. 3), we suggest that locust phase change is less likely to impact presynaptic receptor sensitivity, but that the observed crowding-dependent modulation is probably mediated by lateral network interactions. All other tested mixture combinations (lacking either the colony compound or appetitive food odor) were not different from the response to their components, reflecting a hypo-additive, or elemental, mixture response[41–43]. The comparison of PN response profiles to the food+social mixture revealed higher mixture synergy in gregarious locusts, which is reflected by both a larger proportion of mixture-specific PNs (Fig. 3b) and synergistic response triplets (i.e., [1 | 1 | 4]] and [1|1|6] in Fig. 5f). The resulting odor profile—constructed from all PNs' response statistics—reliably separates gregarious and solitarious phenotypes (LDA analysis, Fig. 5h) and gives rise to specific differences in mixture processing. These predominantly rely on a higher proportion of PNs subjected to antagonistic mixture interactions in solitarious animals and synergistic in gregarious (Fig. 5f). Mechanistically, given the lack of observed excitatory local interneurons (LNs)[44], these changes in network processing are likely mediated by disinhibition of lateral inhibitory LNs, innervated by *Laa*- or *Lct*-sensitive PNs. Local disinhibition will enable motif switches via the temporally and structurally diverse nature of LN populations[45–47] and thus give rise to differential response profiles with increasing population density.

State-dependent regulation of odor processing in the insect antennal lobe has been predominantly investigated in the context of hunger, which incites animals to approach odor cues they would typically avoid (reviewed in refs. 48,49). This reconfiguration involves both modulation of presynaptic receptor sensitivity[50] and recruitment of local interneuronal input by neuromodulatory pathways[46,51].

Another relevant type of context-dependent modulation in olfactory circuits is enhanced courtship receptivity in flies after detecting a food source nearby[52,53]. This enhancement is mediated by lateral excitatory input to the pheromone-sensitive glomerulus in the presence of vinegar smell[53]. In that context, differential lateral processing can be locally modulated by state-dependent (e.g., sex-specific, or virgin vs. mated) modifications to LN innervation patterns[53]. In locusts, similar network-spanning adjustments could occur with changes in population density to enhance food detection in swarms. This can be mediated via adaptive regulation of various components of the interweaved AL network—from local gain control by presynaptic interglomerular inhibition that attenuates ORN-PN signal transfer[47,54,55] to lateral interactions between LN pairs that sharpen odor discrimination and tune transient PN synchronization patterns[56–58]. Similarly, local inhibitory circuits within the mouse olfactory bulb—mediated by dendro-dendritic synaptic connections between mitral cells and granule cells[59,60], analogous to antennal lobe PNs and LNs, respectively—facilitate the discrimination of closely related stimuli[61]. Taken together, nonlinear processing steps are essential for a context-dependent regulation of the olfactory system[62]. However, future research is needed to address how the locust AL network can be fine-tuned for swarming.

Our functional imaging approach complements previous work on locust olfaction characterizing odor mapping as a dynamic code of temporally evolving PN response trajectories that code odor identity[21,22,24,25] in a history-, and context-dependent manner[24,27,63,64]. Now, our study expands on this by mapping the spatial arrangement of odor responses across PNs in a single preparation. We demonstrate that individual odors activate broad, overlapping sets of multiglomerular PNs across the AL. Similar to other insects, these subsets are odor-specific and stable across repetitions (Fig. 4a–c, in agreement with[25]). Recent observations from clonal raider ants—another insect species with largely arborized ALs, similar to locusts—revealed a highly stereotyped odor encoding logic in the ORN population[65]. The many-to-many wiring map of single sensory neurons innervating multiple glomeruli in locusts predicts a broad combinatorial PN response (as seen here), even for narrowly tuned receptors[66]. The implications of such a wiring scheme on coding capacity, which is generally enhanced in the postsynaptic PN layer[67–69], need further investigation.

Furthermore, we recorded from different cellular compartments —PN cell bodies (Fig. 2) and glomeruli (Fig. 4d–h)—revealing synergistic interactions for both. Previous studies suggest that locust cell bodies match the overall PN activity profiles generated via local field potential recordings ([22]. This is in line with PN cell bodies in *Drosophila* clearly exhibiting the odor tuning observed in the respective glomeruli[70]. Yet, our recordings also show differences between somatic and glomerular response properties that potentially reconcile disparities between the literature on locust olfaction and other insects. On the one hand, prolonged and temporally-varying somatic odor responses (Fig. 2f and h) align with electrophysiological recordings from PN cell bodies in locusts[25,40,63]. On the other hand, fast rise and decay times in the glomerular calcium signals (Fig. 4e and h) are consistent with more transient combinatorial maps described in other insects with functional imaging at the dendritic terminals (e.g., refs. [71,72]). Taken together, these points advocate for stable glomerular odor separation across species and AL structures, which is reformatted via network processing into temporally dynamic odor maps for behaviorally relevant representations.

Swarms of desert locusts can span several hundred square kilometers and threaten the food security of millions. The density-dependent phase change from one phenotype to another is accompanied by substantial changes in the quantity, quality, and type of information animals can gather. Our results suggest that crowding already operates in early sensory processing centers, modulating—in our case—the antennal lobe output to influence animals' foraging decisions. The higher proportion of mixture-specific PNs to both food

+social but also food+food mixtures in gregarious animals (Fig. 3b) may indicate a more general adaptation of gregarious locusts to handle multiple simultaneous odor signals in complex environments, such as swarms or varying ecological settings. Yet, to fully understand locust phase change and swarm formation, future research is needed to address the multimodal nature of this phenomenon. Our Bayesian decision-making model suggests that the integration of olfactory with visual information is essential for reliable decision-making, offering new research venues in the fields of vision and multimodal cue integration.

Furthermore, our robust and data-driven calcium imaging analysis provides a comprehensive way to measure odor representation across the locust AL. Given the natural gradient of sociality in locusts, this can be further extended to investigate the cellular mechanisms of adaptive social behavior in changing sensory environments. Altogether, considering their behavioral and social plasticity, locusts provide an excellent opportunity for investigating olfactory processing in complex environments and offer a trackable system for linking sensory adaptations to behaviorally relevant tasks.

## Methods
### Animals
All experiments were performed on gregarious and solitarious desert locusts *Schistocerca gregaria* (Forskål, 1775) obtained from our breeding colony at the animal facility of the University of Konstanz (Konstanz, Germany). Behavioral experiments were done with the last larval instar and functional imaging with freshly molted (<7 days old) adults. Gregarious locusts were reared in crowded cages (~200 animals per $50 \times 50 \times 80$ cm cage), while solitarious locusts were raised in individual boxes ($9 \times 9 \times 14$ cm) with opaque walls in a well-ventilated room with constant air exchange (following[73]). Except for animal density, all other conditions were kept similar for the two colonies with a 12:12 h light:dark cycle, temperature of 27–29 °C, relative humidity of 45% and a diet of in-house grown wheat seedlings, and freshly collected blackberry leaves (*Rubus* sect. *Rubus*). For all experiments, locusts of both sexes were used after one night of starvation. All experiments were carried out following the recommendations of the "3Rs" principles as stated in the Directive 2010/63/EU and all procedures were performed according to the guidelines of the German animal welfare law.

### Behavioral choice assay
We initiated our study with an analysis of the interactions between social and food cues in a patch selection assay (modified from[74]) which presented single locusts with four options: food (blackberry leaves, denoted as *Lvs*), a social setting (eight gregarious locusts, *Lct*), a combination of leaves and locusts (*LvsLct*), and an empty control patch (*Ctr*). Locusts were tested in a 90 cm diameter arena with the four patch choices distributed equally (exchanging positions between trials) in the arena, with a distance of approximately 7 cm from the wall. The four patch choices were made from identical circular plastic containers (Fig. 1a; width: 12 cm; height: 9 cm) and contained either 5 g of ripped blackberry leaves, eight locusts, eight locusts with 5 g of ripped blackberry leaves, or an empty control. In order to allow/prevent access to olfactory and/or visual cues, containers were either transparent with small holes, opaque with holes, or transparent but fully odor-sealed with Parafilm sealing film (Sigma-Aldrich, St. Louis, MO, USA). Each trial lasted 30 min and was recorded at 25 fps by a Basler camera (acA2040−90 $\mu$m; Basler AG, Ahrensburg, Germany) with an IR longpass filter at the top of the arena. Uniform illumination was composed of a ceiling LED ring light (Hakutatz LED ring light, 33 cm, 35 W Bi-color 3200-5600 K) and a grid array of IR 850 nm with a diffuser plate below the arena. In order to prevent potential surrounding visual cues from the lab, uniform white walls of 1 m height were placed around the arena. Experiment containers and the arena

were cleaned after each trial. The ambient temperature during experiments was kept between 24 and 26 °C.

## Analysis of behavioral data

We obtained trajectories of individual locusts using a custom-written GUI[75] in MATLAB (R2023a, The MathWorks Inc, Natick, MA, USA) for automated tracking with manual supervision. Obtained trajectories were temporally smoothed by convolution with a Gaussian kernel (half-width = 2 s, $\sigma = \frac{2}{3}$ s). Visit density at each patch option was estimated as previously described[10] by binning $x/y$-coordinates in a $1000 \times 1000$ pixel grid, followed by a frame-wise application of a 2-D Gaussian smoothing kernel (s.d. = 78, spatial filter domain, filter size = grid size minus one; matching the density-independent interaction range employed by locusts of ~7 cm[76]). The resulting visit density maps were normalized by frame count, divided into four quarters, centered around each patch, and realigned so trials could be pooled across different patch configurations.

To dissect the processes underlying locust patch choice and to evaluate the contributions of the four information classes in the current assay (socio-visual, food-visual, socio-olfactory, and food-olfactory), we extended a Bayesian decision-making rule previously used in patch choice studies (Eqn. (1); refs. 10,29). Given the availability of each information class (e.g., $w_{(socio-olfactory)} = 1$ for options containing locusts in the Olf(+) conditions, but 0 for the control option), we fitted the corresponding reliability parameter ($C_i$) to the experimental visit density data in each patch quarter ($P_x$).

$$P(x\ is\ good|C_i) = \frac{1}{1 + C_i^{-w_j}}$$

$$P_x = \frac{\Pi_{i=1}^{4}P(x\ is\ good|C_i)}{\Sigma_{j=1}^{4}\Pi_{i=1}^{4}P(x\ is\ good|C_i)} \tag{1}$$

We used Bayesian hyperparameter optimization (MATLAB built-in function *bayesopt*) to fit the parameters in the range of $1e^{-1}$ and $1e^{1}$ (log-transformed). We repeated the fitting process 5000 times, resulting in a predictive distribution based on which we estimated 95% credible intervals. Moreover, the nature of parameter $C_i$ allowed us to estimate an information reliability score ($log_{10}(C_i)$) that assesses the degree of attraction (values approaching 1) or aversion (values approaching -1) each information class had on the tested locusts.

## Preparation of animals for functional fluorescence microscopy

We performed functional fluorescence microscopy in the antennal lobes (ALs) of young adult desert locusts by modifying protocols for backfilling AL projection neurons with dextran-coupled calcium indicators[77,78]. For dye application, we anesthetized animals with $CO_2$ to facilitate handling, removed wings and legs, and restrained the locusts in a custom-made holder using dental wax. On the first day, we opened a small window in the head capsule between the compound eyes, exposing the injection site. Next, using a glass capillary, we injected a crystal of a calcium indicator (Cal-520-Dextran Conjugate MW 10'000, AAT Bioquest | Biomol GmbH, Hamburg, Germany) into the medial calixes of both mushroom bodies for a retrograde backfill of AL projection neurons. Following the injection, the brain was covered with locust saline[21], and the head capsule was sealed with a drop of Eicosane (Sigma-Aldrich, St. Louis, MO, USA) to prevent brain desiccation. We allowed the dye to travel overnight with the animal kept in a humid box at 15 °C. On the following day, we exposed the brain and especially both antennal lobes. However, as the antennal base is located directly above the antennal lobes and consequently occluding them, we carefully relocated each antenna ventrally toward the ipsilateral compound eye. Special care was taken to prevent damage to the antennal nerves. Following a successful relocation, we carefully removed the neural sheath covering the AL using fine forceps. For better optical accessibility, we gently elevated the brain with a

stainless steel needle. As a last preparation step, we covered the preparation site with a transparent two-component silicon (Kwik-Sil, WPI, Sarasota, FL, USA).

## Neuroanatomical staining

For morphological visualization of the olfactory pathway, we double-labeled PNs and ORNs using retrograde labeling. PN labeling followed a similar dye loading preparation as for the calcium imaging (see above) using the fluorescent tracer Alexa Fluor 568 conjugated with 10 kDa dextran. For receptor neuron tracing, the antennae were ablated close to the base, exposing the antennal nerve. The exposed antennal nerve was inserted into a glass capillary, loaded with 10 kDa dextran-conjugated Alexa Fluor 488 (both dyes by Thermo Fisher Scientific, Waltham, MA, USA). Animals were incubated for 24 h at 15 °C, after which brains were quickly dissected and fixed in 4% paraformaldehyde (PFA) in phosphate-buffered saline (PBS) for 3 h at 4 °C in the dark. Brains were further washed in PBS and dehydrated in ascending concentrations of ethanol (50, 70, 90, 98, 100%), cleared in xylene for 10 min, and mounted in DPX mounting medium (Sigma-Aldrich, St Louis, MO, USA). Successfully stained antennal lobes were scanned as z-stacks with a Zeiss laser scanning microscope 880 equipped with an Airyscan fast detector and a 25x water immersion objective (LD LCI Plan-Apochromat 25x/0,8 Imm Korr DIC, Carl Zeiss, Jena, Germany) in the Bioimaging Centre at the University of Konstanz.

## Confocal laser scanning microscopy

Confocal microscopy of separate optical sections was acquired using an LSM 510 laser scanning confocal microscope (Carl Zeiss, Jena, Germany), equipped with a $20 \times$ water immersion objective (Zeiss W Plan-Apochromat $20 \times /1,0$ DIC VIS-IR, Carl Zeiss, Jena, Germany). The calcium-sensitive dye was excited with a 488 nm wavelength laser, and emitted light was filtered through a 500–550 nm band-pass filter. Optical sections were acquired at 1 Hz in two different depths ($\sim 35\ \mu m$, and $140\ \mu m$ from the surface of the AL), with a resolution of $0.77\ \mu m/px$ ($x, y$), and a full-width half maximum of $25\ \mu m$ ($z$).

## Odor preparation, olfactory stimulation, and chemical analysis

Odor stimulation was delivered via a custom-built olfactometer with a flow speed of 10 mL/s, and odors were injected for two seconds via a Teflon tube (inner diameter, 0.87 mm) to the ipsilateral antenna[79]. Stimuli were delivered by activating valves that redirected air towards a vial containing 200 $\mu$L of diluted odor. In the case of a single odor, the flow via the odor headspace (200 mL/min) was compensated by closing a balancer that reduces the airflow by the same amount. For mixtures, the headspaces of the two odors (100 mL/min each) were passed through a common Teflon tube, allowing them to mix before arriving at the locust antenna. A 2-min inter-stimulus interval of clean air was used to flush any residues of odors and to let the PNs return to their resting state. The flow of the individual odors and the mixtures was monitored by a photoionization detector (Mini-PID model 200A, Aurora Scientific Inc, Ontario, Canada), which was placed at the opening of the odor delivery tube.

All synthetic compounds (purchased from Sigma-Aldrich, Steinheim, Germany) were diluted in mineral oil. Locusts were tested with the following odorants: cis-3-Hexenyl-acetate diluted to $10^{-2}$ "*Laa*", 3-Hexen-1-ol diluted to $10^{-2}$ "*Z3hl*", 3-Octanol diluted to $10^{-2}$ "*Oct*", and 1-Hexanol diluted to $10^{-3}$ "*Hex*". Odor dilutions were chosen after an initial screening for average PN response magnitude comparable to that of the blackberry leaves *Lvs* in gregarious animals.

Lacking clear dominant candidates for emitted volatiles in the desert locusts (see latest summary of odor composition in ref. 80), we used an extraction from our locust colonies as the social "*Lct*" odor. This was done by placing an evenly spread cotton wool layer in a gregarious colony cage with nymphs of the last larval instar and immature adults for two consecutive days. Odor vials were filled with

either 0.5 g of colony cotton wool or 200 μL of odor dilution on 0.5 g of pure cotton wool. Mixtures were prepared by combining half the amount of the respective odor dilution, leaf extract, or colony cotton wool.

Volatile compounds for chemical analysis of the blackberry leaves were collected via 1 h leaf extract exposure of solid phase micro-extraction (100 μm Polydimethylsiloxane Coating SPME Fiber Assembly, Supelco, Bellefonte, USA) and analyzed with a TRACE GC Ultra Multi-channel gas chromatograph, coupled to DSQ II single quadrupole MS (Thermo Fisher Scientific, Waltham, MA, USA). The leaf extract was produced by grinding 1 g blackberry leaves in 200 μL mineral oil.

### Analysis of calcium imaging data

For our calcium imaging analysis, we obtained a three-dimensional stack (*x,y*, time) for each stimulus and processed the data in MATLAB (Fig. 2d). First, we filtered raw images spatially to account for shot (photon) noise, using a combination of a median filter (3-by-3 pixels window) and a 2-D Gaussian filter (s.d. = 1). Potential bleaching was accounted for by subtracting a trendline (linear polynomial curve fit to each stimulus' median pixel intensity time course). Next, we aligned frames belonging to one stimulus using a non-rigid motion correction algorithm (NoRMCorre[81]) and a 2-D alignment based on phase differences[82]. Carefully aligning our data allowed us to apply a box-shaped kernel (3 frames) in the temporal domain before calculating relative fluorescence changes ($\Delta F/F_0 = \frac{F - F_0}{F_0}$) with the mean pre-stimulus activity five frames before stimulus onset as baseline activity $F_0$. In the case of two repetitions, preprocessed stimulus sets were registered based on phase differences between the sets and eventually averaged.

Interested in an unbiased evaluation of AL responses and detailed quantification of complex response patterns, we referred from manually applying circular regions of interest. Instead, we opted for segmentation into individual, activity-dependent regions (granules) based on the Voronoi topology of the standard-deviation-projection (s.d.-projection) across the whole stimulus set. For this, we identified regional maxima and minima (pixel neighborhood was defined as pixels adjacent in the horizontal, vertical, or diagonal direction) in the s.d.-projection across all stimulus sets. Since the Voronoi diagram and Delaunay triangulation are geometric duals of each other, we could compute Voronoi granules from the Delaunay triangulation of the obtained regional extrema. This resulted in a rough 2-dimensional segmentation into regions that showed substantial changes in activity for any of the stimuli (regional maxima), separated by granules that lacked strong changes (regional minima).

Next, to obtain a more refined segmentation, we evaluated the activity of pixels along the Voronoi vertices. Specifically, we calculated the root median squared error between the activity of the pixel in question and the average activity of the neighboring Voronoi granule or the pixel's 'home' Voronoi granule, respectively. If the pixel had a smaller error with the neighboring Voronoi granule, we assigned it to this area in the next iteration. Alternatively, we kept the original assignment. This evaluation was applied to all pixels along borders between two granules and over a maximum of 50 iterations. Granules smaller than a minimum size of 5 px were fused with their neighboring granules. Last, we removed regions outside the AL and classified all valid ones as 'active' or 'non-active' given each stimulus. For this, we applied Otsu's method to the respective mean intensity projection.

### Construction of odor response vectors

To investigate network-spanning combinatorial coding patterns across all projection neuron cell bodies, we created a response magnitude matrix including data from all gregarious and solitarious animals. In this matrix, each row represents a single PN, and each column corresponds to the respective mean response magnitude during the window of analysis to each of the stimuli (*Laa*, *Lct*, *LaaLct*, *Z3hl*, *Z3hlLct*,

*Oct*, *OctLct*, *Z3hlLaa*, and *OctLaa*; Fig. 2c). To account for inter-animal variability, we normalized the data belonging to a single animal. For this, we initially subtracted the minimum value in order to root-transform the data. Next, we centered the animal's data to have a mean of zero and scaled them to have a standard deviation of one. As a last step, we rescaled the range of each PN's vector (each row) to [-1, 1] before we projected the normalized response matrix into principal component (PC) space, retaining PCs that cumulatively explained at least 99% of the variance.

Following the preprocessing steps, we applied *k*-means clustering to construct distinct odor response vectors. The optimal number of clusters, *k* (1 < *k* ≤ 20), was ascertained by identifying the elbow point of the Variance Ratio Criterion (VRC, Eqn. (2)) curve, which was exponentially decaying with an increasing number of clusters. The VRC is defined as

$$VRC(k) = \frac{SSB(N - k)}{SSW(k - 1)} \qquad (2)$$

Here, *SSB* denotes the between-cluster variance, *SSW* denotes the within-cluster variance, and *N* denotes the number of observations corresponding to the rows in the response matrix. We defined the elbow point as the point on the VRC curve that was farthest from the line segment connecting the first *VRC*(*k* = 2) and last *VRC*(*k* = 20) points on the curve. We performed the *k*-means clustering based on squared Euclidean distances, with a maximum number of $10^6$ iterations, 500 replicates, and with an online update phase in addition to the batch update phase, resulting in six odor response vectors.

### Analysis of response consistency

Consistency in spatial odor representation was estimated with three general odorants: *Laa*, *Hex*, and *Oct*. Responses were captured in the focal depths of the PN cell bodies and glomeruli with repeated stimulus blocks (each stimulus presented once in semi-random order and repeated five times) to estimate the similarity between response maps. For this analysis, the rescaled (range set to [0, 1]) mean intensity maps from each animal were projected into the principal component (PC) space, treating each map as an observation and considering all pixels as features. We retained all PCs that cumulatively accounted for at least 95% of the variance. To evaluate the similarity between maps in PC space, we calculated the Euclidean distance between maps of the same odor stimulus and across maps from different stimuli. Using the average linkage method, we created a hierarchical cluster tree for each animal. Further, we estimated the overall within- and between-stimulus Euclidean distances to pool results across all animals.

### Analysis of dynamical odor response motifs for locust phenotype prediction

To gain deeper insight into the dynamics of odor-induced PN responses and their variation across different stimuli, we identified distinct temporal response motifs. For this, we adopted an approach comparable to the one used to analyze odor response vectors (see above). We used the PN cell body confocal microscopy dataset, encompassing data from all gregarious and solitarious animals, to construct a response profile matrix (Fig. 5a). Unlike the response vectors matrix, we treated all frames as features, each cell body appeared as three observations—one for each stimulus (*Laa*, *Lct*, and *LaaLct*)—and we z-scored each row. However, the principal component analysis-based preprocessing and *k*-means clustering remained consistent, identifying six odor response motifs based on the distinct patterns observed.

Next, we tracked how PN response motif assignments changed with odors. For this, we estimated the transition probabilities between response motifs (e.g., motif 2 → motif 4) from one stimulus to another

(e.g., *Lct → Laa*) for each individual animal. We used this information to fit a discriminant analysis classifier to differentiate between gregarious and solitarious animals based on the dynamics of motif transitions (same diagonal covariance matrix for both phenotypes). The classifier was trained using a leave-one-out approach, with each iteration including all but one animal for training (the model's weights were adjusted to compensate for unequal sample sizes across classes) and employing the model to predict the phenotype of the excluded animal. This procedure was repeated for each animal in the dataset, providing a robust evaluation of the classification accuracy. Further, given the high feature dimensionality, we initially reduced the feature set (transition probabilities) using PCA, retaining all PCs that cumulatively accounted for at least 83% (based on screening different values) of the variance.

## Analysis of synergistic stimulus interaction

To dissect odor response interactions, we analyzed motif triplets using the Bliss score (Eqn. (3)), a standardized measure to categorize pairwise interactions[33].

$$f_{expected} = \left(\frac{f_1}{2} + \frac{f_2}{2}\right) - \left(\frac{f_1}{2} \cdot \frac{f_2}{2}\right)$$
$$f_{observed} = f_3 \tag{3}$$
$$Bliss\ score = 100\left(f_{observed} - f_{expected}\right)$$

Here, $f_1$ and $f_2$ denote the individual response strengths ($\Delta F/F_0$ as decimal rather than percentage) to *Laa* and *Lct*, respectively, while $f_3$ represents the observed response strength to the mixture (*LaaLct*). Positive Bliss scores, which arise when the observed response strength (*LaaLct* response) exceeds the expected response strength, indicate synergistic interaction effects between *Laa* and *Lct*. Conversely, negative values represent antagonistic effects, and values approximating zero indicate additive effects. Note that we divided $f_1$ and $f_2$ by two in Eqn. (3) to account for using only half of each component (*Laa* and *Lct*) when creating the mixture.

## Statistical analysis

All statistical analysis steps were conducted in MATLAB. Unless stated otherwise, we report averages as grand means, which represent the mean of animal means, together with bootstrapped 95% confidence intervals (with a bootstrap sample size of 5000) as shaded areas or error bars, respectively. Further, we used bootstrap randomization tests with $B = 10^7$ samples (cf.[83]) for statistical inference. We employed the following test statistic (Eqn. (4)) for a paired, two-tailed comparison of observations $z$ and $y$ with the resulting pairwise difference $x$ (mean $\bar{x}$, standard deviation $\sigma_x$, sample size $n$)

$$T_{paired\ two-sample} = \frac{|\bar{x}|}{\sigma_x/\sqrt{n}} \tag{4}$$

For an unpaired, two-sample, two-tailed test, comparing observations $z$ (mean $\bar{z}$, standard deviation $\sigma_z$, sample size $n$) and $y$ (mean $\bar{y}$, standard deviation $\sigma_y$, sample size $m$), we used the following test statistic (Eqn. (5))

$$T_{two-sample} = \frac{|\bar{z} - \bar{y}|}{\sqrt{\frac{\sigma_z^2}{n} + \frac{\sigma_y^2}{m}}} \tag{5}$$

For both tests, we applied the same statistical summary to the permutated test statistic and to the test statistic of the original sample to determine the probability $p$ of observing the permutated test statistic being the same or more extreme than the test statistic of the original sample. Note that this approach yields the lowest achievable $p$-value of $1/B = 1/10^{-7}$. In the case of a multiple comparison, the $p$-value was adjusted using the Bonferroni method.

## Reporting summary

Further information on research design is available in the Nature Portfolio Reporting Summary linked to this article.

## Data availability

Source data is provided as a source data file under accession code https://doi.org/10.5281/zenodo.10931394. Source data are provided with this paper.

## Code availability

All analysis code used for this paper is publicly available under accession code https://doi.org/10.5281/zenodo.11190013.

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

## Acknowledgements

The authors thank Yvonne Hertenberger and Hannes Kübler for help with data collection; Nina Schwarz for providing locust illustrations; the team of the Bioimaging Centre at the University of Konstanz for help with obtaining confocal microscopy scans of antennal lobe double stainings; Marco Paoli for valuable insight on calcium imaging and corresponding analyses; Giovanni Galizia for providing feedback and facilities; Dieter Spiteller for GC advice and Hanna Schnell for odor preparation and support with the GC operation and analysis. James Foster, Ahmed El Hadi and Katrin Vogt for insightful comments on the manuscript. This work was completed with the support of the Deutsche Forschungsgemeinschaft (DFG, German Research Foundation) under Germany's Excellence Strategy—EXC 2117–422037984 (supporting Y.G., S.S., S.K. and E.C.F.) and DFG project grant CO 1758/3-1 (E.C.F.).

## Author contributions

I.P., S.S., and E.C.F. established the calcium imaging technique; Y.G. and E.C.F. conceptualized the project and wrote the first manuscript draft; I.P. and S.K. conducted behavior experiments, anatomy, chemical analysis, and functional imaging with S.S. and Y.G. providing conceptual and technical advice; Y.G. recorded EAG data, developed analysis models, analyzed data and created all figures. All authors contributed to writing the final version.

## Funding

## Competing interests

The authors declare no competing interests.
