## [Peer Review File · Nature Communications]

REVIEWER COMMENTS

Reviewer #1 (Remarks to the Author):

The authors report on factors that drive odor guided behavior in desert locusts, which occur in two distinct morphs depending on environmental conditions. Crowding typically induces switching from a solitary morph to a gregarious morph, which is key to swarming under natural conditions, when the locusts can cause significant damage to the natural and agricultural environments. What causes the switch, and the physiological factors that drive it, are therefore important both from the standpoints of basic research as well as agricultural research programs. Thus this work is likely to be of widespread, general interest.

The authors investigate how two factors in particular – visual and olfactory cues – interact in locusts in these two morphs. First, in well planned and executed behavioral experiments, the authors show that solitary phase locusts are drawn primarily to leaf volatiles that represent food whereas gregarious phase locusts are drawn to volatiles combined with the visual cues of other locusts. These experiments point to the potential synergism of cues as they are processed in the brain.

The authors then use calcium imaging of activity in the antennal lobes to evaluate the potential for differential processing of odors in brains of locusts in each of the phases. By imaging a subset of glomeruli (one optical section) the authors show how food and locust-bound social odors interact differently to affect the representation of odors in the antennal lobe. Specifically, there is more excitatory interaction between these odors in gregarious locusts than in solitary, and there may be mixture suppression to some extent in the latter. Moreover, there are more mixture specific PNs for the food-social odor mix in gregarious locusts. CAVEAT: the gregarious/solitary difference for the food-food mix in 3b looks to be almost significant (line 167, $p \sim 0.1$). Would that affect the interpretation. Maybe not. But it might be worth mentioning in the text.

There are two criticisms:

First, any difference in responsiveness in a given brain region between categories of animals – gregarious/solitary in this case – could arise in the local processing in that brain region, as implied in the discussion (lines 274-278). And that, of course, is very important for understanding how the circuits in that brain region are affected by state. But it is also important to rule out changes at earlier processing stages, such as in the antennae. The authors use some reasoning in the lines just cited to argue that the effects they report must be due to changes in antennal lobe circuitry. But it seems like a very circumstantial, and not completely convincing, argument.

Is it too big of an ask at this point to do EAG recordings from antennae of gregarious and solitary animals to establish where stronger responses to components vs mixtures, or mixture synergism, show up in none or the other morph? The recordings are relatively easy to set up as long as the morphs are available. Have these kinds of recordings been done previously by the authors or others? Has anyone looked at expression of ORs/IRs in antennae of the different morphs? Again, one could not establish with expression patterns whether receptors to specific odors were being regulated. But if there were no differences in expression it would boost the argument made in the discussion.

Second, many publications on olfactory responses in the antennal lobes of other insects show temporal patterns. In fact, a lot of information can be gleaned from temporal patterns. Typically, a two- or three-dimensional plot of PC-space shows the evolution of the response over time. This would be a good addition to consider for supplemental materials.

Reviewer #2 (Remarks to the Author):

The goal of the study from Petelski et al. was to determine if there are differences in population level odor encoding in the AL of gregarious vs solitary locusts of ethologically relevant odors. The authors used a robust approach to look at population level responses that were reliable and odor specific, and they demonstrate that differences in metrics of odor representation between behavioral phases are consistent with differences in odor-guided behavior between the behavioral phases. These results are significant, because they provide a neural correlate for differences in perception based on social signals. This study succeeds in its primary goal so my comments revolve primarily upon providing clarification about approaches or caveats.

Specific comments.

-The behavior assay introduced at the onset of the results is novel, so it should have a bit more explanation in the results section and a separate schematic diagram in Figure 1 to help the reader. I had to jump between the results and methods to fully understand how the visual vs. olfactory and odor options were presented, when a simple diagram could have clarified.

-The authors should clarify that while they may have filled all PNs, they likely did not record from all PNs. The language can simply be adjusted to either say "a large subset of PNs" or the authors could calculate how many PNs were recorded in each animal and determine an approximate fraction of PNs recorded based on the number of PNs known to be present in each AL. Personally, I'd just change it to read "a large subset of PNs".

-There was a recent paper from Markus Knaden's group (Chang et al, 2023) demonstrating that locust chemoreceptive proteins are more narrowly tuned relative to those of other insect species. If possible, it would be interesting to discuss the results from this study in which populations of PNs are responsive to many odors with the findings about tuning of chemoreceptive proteins. Because single locust OSNs project to many microglomeruli, it provides a contrast to other insect systems which (based on the present study) have more narrowly tuned PNs. That being said, if there isn't an organic moment in which to discuss this new paper, then the authors shouldn't bother trying to shoe-horn it into their discussion.

-Can the authors comment on the caveats of recording and analyzing odor evoked Ca²⁺ transients from PN somata, since insect neurons are unipolar and the cell body doesn't integrate synaptic input? I understand that to distinguish the activity of individual cells, this was the approach that needed to be taken, but it may be worth commenting on any studies that have compared such transients to electrophysiological recordings.

-The authors should clarify at what point in time odor stimulation occurs for any figure showing odor-evoked responses, such as Figure 1F. They depict the time at which the analysis was performed, but not when the stimulus was provided.

-The authors should provide greater explanation of how the Bliss score over time was calculated. I got caught up in Figure 5E (the cartoon representation of the analysis of the triplet motifs) because it starts by showing Ca²⁺ transients to represent each odor-specific motif type and then transitions into Bliss scores over time. Further clarification would help ensure that the reader can follow the logic of how this analysis was implemented.

-The authors could beef up their citations for labs demonstrating gain-control by lateral inhibitory interactions within the AL. Currently, it is pretty sparse, but quite a few groups have shown gain control regulation by local interneurons across a variety of species

RESPONSE TO REVIEWERS' COMMENTS

Reviewer #1 (Remarks to the Author):

The authors report on factors that drive odor guided behavior in desert locusts, which occur in two distinct morphs depending on environmental conditions. Crowding typically induces switching from a solitary morph to a gregarious morph, which is key to swarming under natural conditions, when the locusts can cause significant damage to the natural and agricultural environments. What causes the switch, and the physiological factors that drive it, are therefore important both from the standpoints of basic research as well as agricultural research programs. Thus this work is likely to be of widespread, general interest.

The authors investigate how two factors in particular – visual and olfactory cues – interact in locusts in these two morphs. First, in well planned and executed behavioral experiments, the authors show that solitary phase locusts are drawn primarily to leaf volatiles that represent food whereas gregarious phase locusts are drawn to volatiles combined with the visual cues of other locusts. These experiments point to the potential synergism of cues as they are processed in the brain.

The authors then use calcium imaging of activity in the antennal lobes to evaluate the potential for differential processing of odors in brains of locusts in each of the phases. By imaging a subset of glomeruli (one optical section) the authors show how food and locust-bound social odors interact differently to affect the representation of odors in the antennal lobe. Specifically, there is more excitatory interaction between these odors in gregarious locusts than in solitary, and there may be mixture suppression to some extent in the latter. Moreover, there are more mixture specific PNs for the food-social odor mix in gregarious locusts.

Response: Thank you for your great suggestions! Including both - EAG experiments and PC-space trajectories over time - helped to improve our manuscript considerably. Please see our detailed responses below.

CAVEAT: the gregarious/solitary difference for the food-food mix in 3b looks to be almost significant (line 167, $p \sim 0.1$). Would that affect the interpretation. Maybe not. But it might be worth mentioning in the text.

Response: Thank you for highlighting this. We interpret our observation that a synergistic interaction between social and food-related stimuli in gregarious animals is matched by an increase in mixture-specific PNs as an adaptation for enhanced food detection in swarms. Now, there seems to be a trend of also more PNs that are mixture-specific for the food-food blend in Fig. 3 - as pointed out - despite the fact that overall response strength for the combination of the two food stimuli is not higher than the individual components (Fig. 2). All together this may indicate a general adaptation for processing multiple simultaneous odor signals in complex environments such as swarms or varying ecological settings. In sum, the observation of an almost significant difference between gregarious and solitary animals in mixture-specific food-food PNs may point towards a more general modulation in mixture processing, but at this

stage, it is less conclusive than the clear synergism observed for the locust+food mixture. We added a short discussion about this in the text (lines 354-356).

There are two criticisms:

First, any difference in responsiveness in a given brain region between categories of animals – gregarious/solitary in this case – could arise in the local processing in that brain region, as implied in the discussion (lines 274-278). And that, of course, is very important for understanding how the circuits in that brain region are affected by state. But it is also important to rule out changes at earlier processing stages, such as in the antennae. The authors use some reasoning in the lines just cited to argue that the effects they report must be due to changes in antennal lobe circuitry. But it seems like a very circumstantial, and not completely convincing, argument.

Is it too big of an ask at this point to do EAG recordings from antennae of gregarious and solitary animals to establish where stronger responses to components vs mixtures, or mixture synergism, show up in none or the other morph? The recordings are relatively easy to set up as long as the morphs are available. Have these kinds of recordings been done previously by the authors or others? Has anyone looked at expression of ORs/IRs in antennae of the different morphs? Again, one could not establish with expression patterns whether receptors to specific odors were being regulated. But if there were no differences in expression it would boost the argument made in the discussion.

Response:

Yes, that's a very good point. As you suggested, EAG was relatively easy to set up, and we were intrigued to test if responses also differ at the periphery/antenna. Using all solitarious locusts left in our colony, we tested 26 solitarious and 35 gregarious antennae, and the answer is: no phenotypic difference on the antenna level.

Interestingly, in both phenotypes, there was a tendency of increased responses to the food+social mixture compared to the food-only stimulus, but response magnitudes are comparable between gregarious and solitarious locusts (and generally in line with literature reporting larger EAG response magnitudes for mixtures [1]), suggesting that phenotypic modulation occurs primarily in the antennal lobe.

Having said that, and in line with your further suggestions, EAG signals are inherently challenging to interpret as they reflect the summed potentials along the whole antenna, this can call for a full future project (even projects) on the single-sensillum level. While it is known that solitarious locusts have more olfactory sensilla than gregarious ones [2] and that there are no differences in the response of coeloconic or trichoid receptor neurons [3], we are unaware of a comparison between expression patterns of ORs and IRs between the phenotypes, especially not for the desert locust (for the migratory locust a large subset of ORs and IRs has been recently annotated [4], but we are unaware of a phenotypic comparison of expression level), but this is a super idea for future work!

For now, we added a new supplementary figure with our findings, concluding that antennal recordings do not point towards social modulation at the antennal level (suppl. Fig. 3, related text lines 142-145 and 288-289, with details in suppl. Methods).

[1] [https://doi.org/10.1016/0022-1910\(84\)90032-5](https://doi.org/10.1016/0022-1910(84)90032-5)

[2] <https://doi.org/10.1007/s004410051022>

[3] <https://doi.org/10.1046/j.1365-3032.1999.00107.x>

[4] <https://doi.org/10.1007/s00018-015-2009-9>

Second, many publications on olfactory responses in the antennal lobes of other insects show temporal patterns. In fact, a lot of information can be gleaned from temporal patterns. Typically, a two- or three-dimensional plot of PC-space shows the evolution of the response over time. This would be a good addition to consider for supplemental materials.

Response: Thank you for the suggestion. We fully agree and added the analysis to the revised version. We calculated an average coefficient matrix based on all animals (all cell bodies from all gregarious and solitary animals) in our dataset. Using this matrix, we projected each animal's data into PC-space to calculate the grand mean (mean of animal means) trajectories for gregarious and solitary animals (see the new subsection in the supplementary material for more details). The analysis revealed that the different stimuli (food, social, and their mixture) describe distinct population trajectories in PC-space (see the new supplementary figure 04). The general directions of these trajectories are similar for gregarious and solitary animals, but differences remain noticeable. Specifically, values for the mixture response in PC1 (explaining 60.4% of the variance) are, for example, more extreme for gregarious animals, whereas PC1 values for the response to the food-related stimulus are more extreme in solitary animals. Reconstructing the signal without PC1 showed that it is strongly linked to the signal magnitude, explaining the differences observed in PC-space. We present and discuss this in the Results and Discussion sections (lines 131-134 and lines 286-287).

Reviewer #2 (Remarks to the Author):

The goal of the study from Petelski et al. was to determine if there are differences in population level odor encoding in the AL of gregarious vs solitary locusts of ethologically relevant odors. The authors used a robust approach to look at population level responses that were reliable and odor specific, and they demonstrate that differences in metrics of odor representation between behavioral phases are consistent with differences in odor-guided behavior between the behavioral phases. These results are significant, because they provide a neural correlate for differences in perception based on social signals. This study succeeds in its primary goal so my comments revolve primarily upon providing clarification about approaches or caveats.

Response: Thank you for the thoughtful feedback. Incorporating your suggestions contributed a lot to the manuscript's clarity and quality, as well as helped us to better link it with a broader relevant olfaction literature. Please see our detailed responses below.

Specific comments.

The behavior assay introduced at the onset of the results is novel, so it should have a bit more explanation in the results section and a separate schematic diagram in Figure 1 to help the reader. I had to jump between the results and methods to fully understand how the visual vs. olfactory and odor options were presented, when a simple diagram could have clarified.

Response: An illustration is a great idea for explaining the tested conditions. We now include the color-inverted, exemplary frame from suppl. Fig. 01 in the main figure and add a diagram showing the different container conditions (V+O+, V-O+, and V+O-). Moreover, we now provide more details in the results section as well (lines 80-82).

The authors should clarify that while they may have filled all PNs, they likely did not record from all PNs. The language can simply be adjusted to either say “a large subset of PNs” or the authors could calculate how many PNs were recorded in each animal and determine an approximate fraction of PNs recorded based on the number of PNs known to be present in each AL. Personally, I’d just change it to read “a large subset of PNs”.

Response: True. We have corrected the text to “a large subset of PNs”, thank you for pointing it out.

There was a recent paper from Markus Knaden's group (Chang et al, 2023) demonstrating that locust chemoreceptive proteins are more narrowly tuned relative to those of other insect species. If possible, it would be interesting to discuss the results from this study in which populations of PNs are responsive to many odors with the findings about tuning of chemoreceptive proteins. Because single locust OSNs project to many microglomeruli, it provides a contrast to other insect systems which (based on the present study) have more narrowly tuned PNs. That being said, if there isn't an organic moment in which to discuss this new paper, then the authors shouldn't bother trying to shoe-horn it into their discussion.

Response: This is a good point that we now included in the discussion (lines 332-336). We also think, as you and the paper by Knaden's group suggest, that response broadness at the PN level results from the complex (possibly largely divergent) wiring of the locust AL. We elaborated our discussion on this.

Can the authors comment on the caveats of recording and analyzing odor evoked Ca²⁺ transients from PN somata, since insect neurons are unipolar and the cell body doesn't integrate synaptic input? I understand that to distinguish the activity of individual cells, this was the approach that needed to be taken, but it may be worth commenting on any studies that have compared such transients to electrophysiological recordings.

Response: Thank you for giving us the chance to elaborate on this. Indeed, as pointed out, analyzing cell body calcium responses is crucial for assigning cell identity in this highly distributed network and enables revealing the integration of microglomerular activity. In the

revised version we addressed the concern about whether calcium levels in cell bodies reflect dendritic activity by referring to the following lines of evidence:

1. Previous intracellular recordings from locust PN cell bodies match overall PN activity profiles generated via local field potential recordings [1]. Such intracellular data has been used extensively for discovering locust antennal lobe functions (e.g. see [2]).
2. Root *et al.* [3] recorded calcium responses in both cell bodies and glomerular levels in the *Drosophila* antennal lobe while using specific *gal4* drivers to match corresponding units. Cell bodies clearly exhibited the odor tuning observed in respective glomeruli. Odor-induced activity levels were also highly correlated.
3. Cell body calcium activity has been investigated in other loci of olfactory circuits: mushroom body Kenyon cells. These neurons are highly branched with overlapping innervation. Several papers successfully recorded cell bodies to analyze cell identity and odor coding [4,5].

Taken together, these references support our approach. We addressed this in the revised discussion (lines 337-341)

[1] <https://doi.org/10.1523/JNEUROSCI.16-12-03837.1996>

[2] <https://doi.org/10.1016/j.neuron.2005.09.032>

[3] <https://doi.org/10.1073/pnas.0704523104>

[4] <https://doi.org/10.1523/JNEUROSCI.1099-11.2011>

[5] <https://doi.org/10.1038/nature15396>

The authors should clarify at what point in time odor stimulation occurs for any figure showing odor-evoked responses, such as Figure 1F. They depict the time at which the analysis was performed, but not when the stimulus was provided.

Response: True. Stimulus onsets are now indicated with vertical lines on the dF/F traces.

The authors should provide greater explanation of how the Bliss score over time was calculated. I got caught up in Figure 5E (the cartoon representation of the analysis of the triplet motifs) because it starts by showing Ca²⁺ transients to represent each odor-specific motif type and then transitions into Bliss scores over time. Further clarification would help ensure that the reader can follow the logic of how this analysis was implemented.

Response: Thank you for pointing this out. To clarify it better, we have added more details to the Results section (lines 230-233) and to the caption of Figure 5.

The authors could beef up their citations for labs demonstrating gain-control by lateral inhibitory interactions within the AL. Currently, it is pretty sparse, but quite a few groups have shown gain control regulation by local interneurons across a variety of species.

Response: True. This is an important direction to expand the discussion, thanks for pointing it out. The revised version now includes a wider overview on gain-control and lateral regulation in the AL network, referring to more work by different groups and organisms (lines 314-323), which we believe indeed strengthens the discussion substantially.

REVIEWERS' COMMENTS

Reviewer #1 (Remarks to the Author):

The authors have responded well to the comments from the reviews. I am happy with the new work they have done and how they responded overall to my comments.

Reviewer #2 (Remarks to the Author):

The authors have addressed all of my concerns, providing further clarification in the text and in figures, as well as addressing some caveats that were needed. I commend the authors for their work and for their willingness to include new data, new analyses and new schematics. This is an excellent study that will be of broad general interest.